# Olfactory and Gustatory Supra-Threshold Sensitivities Are Linked to *Ad Libitum* Snack Choice

**DOI:** 10.3390/foods11060799

**Published:** 2022-03-10

**Authors:** Sashie Abeywickrema, Rachel Ginieis, Indrawati Oey, Mei Peng

**Affiliations:** 1Sensory Neuroscience Laboratory, Department of Food Science, University of Otago, Dunedin 9054, New Zealand; sashie.abeywickrema@postgrad.otago.ac.nz (S.A.); rachel.ginieis@postgrad.otago.ac.nz (R.G.); indrawati.oey@otago.ac.nz (I.O.); 2Riddet Institute, Private Bag 11 222, Palmerston North 4442, New Zealand

**Keywords:** sensitivity, discriminability, olfaction, gustation, food choice, snacking, obesity

## Abstract

Snacking is a common eating habit in the modern food environment. Individual snack choices vary substantially, with *sweet* versus *savoury* snacks linked to differential health outcomes. The role of olfactory and gustatory sensitivities in snack choices and consumption is yet to be tested. A total of 70 Caucasian young males (age: 21–39 years; BMI: 20.5–40.5 kg∙m^−2^) were tested for their supra-threshold sensitivities to *sweet* and *savoury* associated odours and tastants (vanillin, methional; sucrose, NaCl). The participants also attended an *ad libitum* task in which their intakes of *sweet* and *savoury* snacks were recorded and analysed. Univariate and multivariate analyses were used to test for relationships between odour/taste sensitivities and *sweet* versus *savoury* snack intake. Results indicated that individual sensitivities to sweet-associated stimuli (e.g., vanillin, sucrose) were negatively linked with intake of the congruent (e.g., *sweet*) snacks and positively linked with incongruent (e.g., *savoury*) snacks (*p* < 0.05). These differences were reflected by energy intake rather than consumption weight (*p* > 0.05). This study outlines the fundamental roles of olfactory and gustatory sensitivities in snack choices and offers novel insights into inter-individual variability in snack consumption.

## 1. Introduction

Over the past few decades, individual snacking frequency and intake have increased substantially, commensurate with the rapid growth of obesity [1]. While consuming low-energy snacks may help to maintain a healthy lifestyle, the current food environment offers an abundance of sensorially appealing yet energy-dense snacks. Despite the omnipresence of snacks in the current food environment, some individuals seem particularly susceptible to choosing “unhealthy” food items [2,3,4]. While extensive research has been conducted to understand inter-individual differences in snacking behaviour, findings to date remain inconclusive. In particular, the potential role of individual flavour sensitivities in shaping snack choice represents an important question yet to be fully addressed.

Despite increasing research attention to problems associated with “snacking”, a consensus on its definition has not been reached. Some studies define snacks according to dietary quality and nutrient composition [5,6], whereas other studies have used a temporal approach by defining snacks as foods or beverages consumed between main meals [7,8]. Over the past few decades, both snacking frequency and variety have increased, although recent studies highlight that snack types and intake, rather than timing and frequency, are more important correlates to weight issues [9,10,11].

Commercial snacks are loosely divided into sweet and savoury categories, subject to their primary sensory characteristics [12,13]. Typically, energy-dense snacks are deemed as “unhealthy” [14], with increasing evidence linking overconsumption of snacks to health conditions such as heart disease and cancer [15,16]. Intriguingly, research has revealed differential health outcomes linked to consumption of high-sweet versus high-salt snacks. Specifically, frequent snacking on savoury foods has been linked to significant increases in blood pressure [17,18], whereas consumption of sweet snacks has been linked to diabetes [19,20].

Individual snack choices can vary substantially, with two key quantifiers—portion size and energy density [21]. While both of these consumption parameters are important, Peng, Cahayadi [22] found that individuals differed more substantially when judging a snack’s energy density than for estimating its portion size. Labbe, Rytz [23] concluded that choice for large portions could be motivated by either perceived healthiness or tastiness of the food. Their findings indicated that portion size alone provided little information and, instead, energy density of selected foods was a better indicative measure for inter-individual difference in eating. Additional research investigating key motivators for snack choices has consistently suggested “sensory appeal” as the most prominent factor [13,24]. Indeed, Cleobury and Tapper [13] reported that 55% of energy-dense snacks were chosen because “*the food looked/smelt tempting*”. These findings highlight the important role of individual sensory perception in orientating snack choices.

The olfactory sense plays a key role in perception of food flavours and for determining the hedonic value of food. However, olfactory sensitivity is also known for its remarkable variability across individuals [25,26], with age, body mass index (BMI), and cultural background all having profound effects on this measure (e.g., [27,28,29,30]). Previous studies suggest that such variability in olfactory perception may contribute to inter-individual differences in food choices. For instance, Jaeger, McRae [31] showed that odour sensitivity to β-ionone can directly determine acceptance of foods containing β-ionone (e.g., raspberry), with higher sensitivities linked to less intake and vice versa. Furthermore, weakened sense of smell has been shown to contribute to weight gain by shifting food choices to energy-dense categories [32,33,34]. In elderly populations, declines in olfactory acuity are often accompanied by enhanced preference for sweet foods [35,36]. A recent study by Ginieis, Abeywickrema [37] tested for links between individual olfactory sensitivity versus habitual and snack intake and observed a negative relationship only with the latter parameter. These results suggest olfactory sensitivity potentially has a stronger influence on snacking than on habitual food intake.

Individual gustatory sensitivities have been strongly linked to food choice, with particular importance for sensory-congruent foods. For instance, a strong negative relationship was observed between sensitivity to fatty taste and preference for high-fat foods [38,39,40]. Similar relationships have been observed for sensitivity to umami, salt taste, and savoury foods and between sweetness sensitivity and sweet foods [40,41,42], but see [43,44] for different results. In addition, a recent study by Han, Keast, and Roura [45] showed that higher sensitivities to sweet taste are associated with a decrease in *ad libitum* sweet food intake. In their experiment, the participants freely selected and consumed food across different sensory categories. Notably, Keast and Roper [46] attempted to link inter-individual differences in taste detection threshold as well as in supra-threshold intensity measures. Their results suggested a significant sensitivity–intake relationship with supra-threshold measures but not for detection threshold. These authors emphasised the importance of using supra-threshold measures for linking sensory function to food intake.

Smell and taste perception are closely intertwined, with extensive evidence for cross-modal interactions [47,48]. A classical example of such an interaction, discussed by Dravnieks, Masurat [49] and Harper, Smith [50], involves specific sweet-smelling odours, such as vanilla. This effect was reported as a phenomenon of associative learning by Stevenson, Prescott [51]. While olfaction and gustation are important for food choices, there is a recent debate regarding the relative functional roles of these two senses in appetite control, food choices, and consumption behaviour. Boesveldt and de Graaf [52], in a systematic review, ascertained that olfaction, versus gustation, is more closely linked to anticipatory rewards, thus playing a more important role in food choice and intake. This hypothesis has been recently tested by contrasting olfactory versus gustatory effects on satiety [53] and food intake [54,55]. It remains unclear, however, whether individual olfactory and gustatory sensitivities play differential roles in orientating food choices.

The present study aims to test for relationships of individual olfactory and gustatory sensitivities and their snack choices in a laboratory-based *ad libitum* setting. This study is the first to systematically assess the role of an individual’s olfactory and gustatory sensitivities in influencing snack choice. Built upon previous research (e.g., [31,42]), we hypothesise that olfactory and gustatory sensitivities are negatively linked to consumption of snacks containing congruent characteristics. Specifically, the study specifically tests for difference across individuals with high and low sensitivities in snack choices in terms of portion size and energy density. Findings from this study offer new insights into links between sensory sensitivities and snacking, shedding light on inter-individual differences in snack choices associated with differential health risks.

## 2. Methods

### 2.1. Participants

A total of 70 Caucasian males aged 20–40 years from the general community of Dunedin, New Zealand were recruited to participate in this study. In order to alleviate variabilities in eating behaviour due to ethnicity, age, and sex, only Caucasian young males were included in the study [56,57]. Additionally, individuals with sensory dysfunction, restraint eating behaviour, food allergies, BMI under 18.5 kg∙m^−2^, or smoking habit were excluded from the study. Informed written consent was obtained from each participant before the commencement of their participation. At the completion of the study, each participant was given monetary compensation. The study had been approved by the University of Otago Human Ethics Committee (Reference number: H18/111).

### 2.2. Overview of the Study

The study was carried out at the Sensory Neuroscience Laboratory of the University of Otago. Each participant attended five, 1-h test sessions between 7:00 a.m. and 9:00 a.m. on consecutive weekdays. Participants were instructed to restrain from food and non-water beverages for at least 10 h before each session. The experiment started with four sessions of sensitivity tests, which tested individual difference thresholds to two odour and two taste stimuli, with each stimulus being tested twice in separate sessions. In the last session, the participants were asked to undertake a laboratory-based *ad libitum* snack choice task, demographic questionnaires, and anthropometric measures. Specifically, each session of the sensitivity tests included two separate tests to one gustatory and one olfactory stimulus. The orders of the sensory tests were randomised across the participants using William’s Latins Square. In the last session, all participants were requested to complete a Dutch Eating Behaviour Questionnaire (DEBQ) [58], which assesses individual tendencies for restrained (1—unrestrained; 5—restrained eaters), emotional (1—unemotional; 5—emotional eaters), and external (1—non-external; 5—external eaters) eating behaviour.

Each participant’s height (in cm) and weight (in kg) were measured at the end of the experiment to the nearest 0.1 unit. Participants were instructed to remove heavy clothes and shoes prior to weighing. These measures were then computed into BMI (kg/m^2^; [59]). The data collection was organised by cohorts of five participants.

### 2.3. Sensory Experiments

#### 2.3.1. Stimuli

Two food-related olfactory and two gustatory stimuli were used for the sensory tests. Details on the stimuli are presented in Table 1. Two odour stimuli, vanillin (O1; vanilla smell) and methional (O2; potato smell) were selected due to their close relevance to snack foods that are common to NZ consumers [60,61,62]. The two gustatory stimuli were sucrose (T1: sweet taste) and salt (T2: salty/savoury taste). These odour and taste stimuli had been demonstrated in previous research to be associated with perception of sweet and savoury foods [63,64].

To perform the sensitivity test for each stimulus, a reference concentration, representing moderate intensity perception, was firstly determined. Subsequently, an incremental concentration series, consisting of five logarithmic steps, was made (see Table 1 for exact concentrations). All sample preparation was carried out using a serial dilution technique, using filtered water (0.5 microns) as the solvent. Solutions were then sub-sampled into 50 mL brown glass bottles and labelled with unique three-digit codes. All samples were prepared beforehand and kept refrigerated. On testing day, samples were diffused to room temperature before serving.

#### 2.3.2. Sensitivity Tests

Each of the sensory testing sessions comprised two tests—one for olfactory and one for gustatory stimuli. Each test included five blocks of method of constant stimulus 2-alternative-forced-choice (2-AFC) presentations. A block comprised five 2-AFC trials, with each trial testing for a specific stimulus step. Specifically, in each trial, the participant was presented with two samples containing a reference sample and a target sample (from the incremental stimulus steps). After tasting (or smelling) the samples in a given order, the participant was asked to indicate which one of these samples gave a stronger taste (or smell). A 1-min inter-trial interval and additional 10-min break between blocks were enforced. Participants were instructed to smell or drink sparkling mineral water to cleanse their palate and to moderate any carryover effect. Compusense Cloud software (Guelph, ON, Canada) was used to design sensitivity assessment experiments and data collection.

Across two replicated sessions, the participant completed a total of 10 blocks, producing hit rates (i.e., H; correctly recognizing the higher concentration) and false alarm rates (FA; mistakenly recognizing the lower as the higher concentration) for each stimulus step. Then, the sensitivity measure, *d*′ values (*d*′ = 1/√2 [z(H) − z(FA)]), were calculated using the formula given [65]. All of the above calculations were performed using Excel (Microsoft Office, 2018, USA).

### 2.4. Laboratory-Based Ad Libitum Snack Choice Task

Individual snack choices were assessed with a laboratory-based *ad libitum* task. Eight types of snack common to New Zealand consumers were selected [66,67], including four *sweet* (i.e., chocolate, chocolate chip cookies, crunchy granola, red apples) and four *savoury* categories (i.e., salted peanut, corn chips, rice crackers, steamed broccoli). Details of these snack items are presented in Table 2.

To control for the baseline satiety at testing, participants received a portion of oat porridge as breakfast. This breakfast portion was customised for each participant to cover 10% of their daily energy requirements, calculated via the Harris–Benedict equation: BMR_(Men)_ = (66.47 + (13.75 × weight in kg) + (5.003 × height in cm) − (6.755 × age in years)) [68]. This calculation was performed based on participants’ self-reported height and weight measures collected during the initial recruitment. At 20 min post-breakfast, participants were asked to report their hunger level on a 100 mm visual analogue scale (VAS) and then individually presented with bite-sized portions of eight different snack items. Participants were firstly asked to rate their liking and wanting for each snack on separate 100 mm VAS scales. This tasting component aimed to introduce the sensory characteristics of the snacks to the participants before they had access to the food buffet.

Subsequently, participants were led into a testing room with the snack buffet and asked to select as many snack items as they would like to eat (e.g., [69,70]). After the selection, the participant returned to an individual booth to eat the snacks while completing a questionnaire. All participants were instructed to keep the snack leftovers on the plate and leave the booth at the end of the session. The food consumed by each participant was calculated by weighing their initial selection and leftovers. The participant was unaware that their consumption was monitored or measured until the full disclosure at the end of the experiment (they were then offered the opportunity to withdraw their data).

Intake of each snack item was first recorded in weight (in g) and then computed into energy measures (in kJ). For analytical purposes, these measures were then grouped into *sweet* and *savoury* categories. Additionally, energy densities of individual choices over *sweet* and *savoury* categories were computed by dividing energy by weight (kJ·g^−1^).

### 2.5. Data Analysis

Descriptive statistics were performed on the demographic, DEBQ, and snack intake data. In addition, a series of paired sample *t*-tests was employed to evaluate the difference between *sweet* and *savoury* snack intake. Analyses were employed for each weight, weight percentage, energy, and energy density measure.

Regarding the main analyses, participants were first divided into “low” and “high” sensitivity groups based on the median split of sensitivities to each of the testing stimuli. Repeated-measures ANCOVA was used to test the differences of the *sweet* and *savoury* snack intake between “high” and “low” *sensitivity groups* for separate stimuli. Each ANCOVA thus included the *snack category* (i.e., *sweet* and *savoury*) as a repeated-measures within-subject variable and the *sensitivity group* (i.e., high and low *d*′ within separate median range) as a between-subject variable. BMI was treated as a continuous covariate during the analysis. Analyses were employed for weight and energy measures. Significant results were further tested by post-hoc tests with *Bonferroni* corrections.

A series of Pearson’s correlation analyses were employed to test the relationship between sensitivity to each olfactory (i.e., O1, O2) and gustatory (i.e., T1, T2) stimulus and energy density consumed from *sweet* and *savoury* snack choices, separately. Data of the present study have been deposited in Open Science Framework, OSF; https://osf.io/tq3ew/ (data has been deposited on 8 February 2022). R was used as the main analytical software (v. 1.1.463, Boston, MA, USA).

## 3. Results

### 3.1. Descriptive Statistics for Participant Characteristics and Snack Intake

Of the 70 participants, one participant’s BMI was deemed an outlier (48.1 kg·m^−2^); thus, data of this participant were removed from the following analyses. Overall, the analyses included data from 69 participants. Table 3 summarises participants’ characteristics.

On average, the participants consumed 159.8 g (±68.5) and 877 kJ (±562) from the snack buffet. Additionally, the averaged energy density of all snack consumption was 5.6 kJ·g^−1^ (±2.0). A series of paired sample *t*-tests was performed to test for differences between sweet and savoury snack intake in terms of weight, weight percentage, energy, and energy density (Table 4). Results indicated that *sweet* snacks were associated with significantly higher measures than the *savoury* snacks, except for energy density.

### 3.2. Comparison of Sweet and Savoury Snack Intake between “High” and “Low” Sensitivity Groups

Repeated-measures ANCOVA was performed on weight and energy measures to test differences between the “high” and “low” sensitivity groups for separate sweet and savoury snack intake. BMI was included in the analyses as a continuous covariate.

Results of the analyses are presented in Figure 1. In terms of the weight measure, none of the repeated-measures ANCOVAs returned any significant results, indicating similar consumptions between high- and low-sensitivity groups for any of the testing stimuli. BMI was not a significant covariate for any of these analyses.

By contrast, repeated-measures ANCOVA on energy measures indicated significant interaction effects for O1 (F_(1,130)_ = 5.03, *p* = 0.026), O2 (F_(1,130)_ = 3.12, *p* = 0.042), and T1 (F_(1,130)_ = 3.21, *p* = 0.047) stimuli. Post-hoc tests with *Bonferroni* corrections were performed to test for differences between sensitivity groups in either *sweet* or *savoury* snack intake. In summary, individuals who were more sensitive to O1 and T1, compared to their low-sensitivity counterpart, consumed less energy from the *sweet* category (O1: *p* < 0.001; T1: *p* = 0.047) but consumed more energy from the *savoury* category (O1: *p* = 0.046; T1: *p* = 0.049). The sensitivity groups for O2 only showed significant differences in the *sweet* intake (*p* = 0.016; *savoury* intake: *p* = 0.174). Analysis based on T2 showed no significant results (F_(1,130)_ = 1.30, *p* = 0.385). BMI, as the covariate, had no significant effect or interaction in any of the models.

### 3.3. Relationship between Odour and Taste Sensitivities and Variabilities in Energy Density of Snack Choices

A series of Pearson’s correlation analyses was employed to test the relationship between *d*′ to each stimulus and energy density of *sweet* and *savoury* snacks separately (see Figure 2). Results revealed that *d*′ to O1 and T1 had a significantly negative correlation to energy density of selected *sweet* snacks (O1: r = −0.36, *p* = 0.009, T1: r = −0.33, *p* = 0.017). By direct contrast, a positive correlation was observed between *d*′ to O2 and energy density of the *sweet* snacks (r = 0.28, *p* = 0.046). Sensitivities to T2 showed no significant relationship with energy density of the *sweet* snacks (r = −0.11, *p* = 0.158). These results did not differ with BMI being controlled (partial correlation coefficients: O1: r = −0.35, *p* = 0.009, O2: r = 0.27, *p* = 0.046, T1: r = −0.33, *p* = 0.017, T2: r = −0.10, *p* = 0.160).

With regards to energy density of savoury snacks, significantly positive correlations were observed for *d*′ of O1 and T1 (O1; r = 0.34, *p* = 0.027, T1; r = 0.27, *p* = 0.040). The *d*′ values for neither O2 nor T2 were significantly correlated with energy density of savoury snacks. These relationships remained similar when BMI was controlled (partial correlation coefficients: O1: r = 0.33, *p* = 0.027, O2: r = −0.17, *p* = 0.067, T1: r = 0.28, *p* = 0.041, T2: r = −0.10, *p* = 0.190).

## 4. Discussion

The present study tested for links of individual olfactory and gustatory supra-threshold sensitivities to snack intake in an *ad libitum* setting. Findings from the study only partially supported our original hypothesis that olfactory and gustatory sensitivities are negatively linked with sensory-congruent snack consumption. Specifically, individuals who were more sensitive to sweet-associated odour and taste cues consumed less energy from sweet snacks, despite selecting similar portions (weight) as their less sensitive counterparts. Additionally, heightened sensitivities to three out of the four testing stimuli were linked to increased energy consumption of sensory-incongruent snacks, suggesting the presence of sensory-compensatory effects.

On average, participants of the present study consumed more *sweet* (61.9%) than *savoury* (38.1%) snacks. These percentages were slightly different from Halford, Boyland [71], who reported 84% favouring sweet and 16% favouring savoury snacks, in a comparable *ad libitum* setting. Previous studies also reported that individuals with elevated body weights tended to show stronger preference for sweet snacks (e.g., [11,72,73,74]). However, the present study failed to detect this relationship, with BMI playing a minimal role in predicting either sweet or savoury snack consumption. These differential findings may be partly attributable to laboratory-based *ad libitum* design (see [75,76]).

Links between taste sensitivities and food choice have been extensively discussed in previous literature, with findings often remaining controversial. Of the different taste qualities, sweetness is relatively well-studied, with both positive and negative relationships reported between sweet sensitivity and sweet food consumption [41,42,43], while some studies report no such difference [44,77]. The present analysis consistently indicates that higher sensitivities to sweetness lead to reduction in energy intake of sweet snacks. Such observations are in line with the sensory compensation effect, which had been proposed by Bartoshuk, Duffy [78] to describe how individuals with high sensitivity to a specific tastant have relatively low preference for foods with congruent tastes. This effect had been demonstrated in various studies based on different measures of sensitivities (e.g., [41,42,43,79]). Notably, a few studies that used detection thresholds did not observe such a relationship [80,81]. Discrepant findings across these studies and the present study highlight the importance of sensitivity measures when detecting sensory effects on food choices.

Relationships between *odour* sensitivity and food choice have received comparatively little attention. Jaeger, McRae [31] remains the only study, to our knowledge, to directly test for olfactory sensitivity to β-ionone and its link to acceptance of sensory-congruent foods. Their study observed a negative relationship between sensory and intake measures, in line with the sensory compensation effect. The present study, expanding from these previous findings, indicates that such effects might be extended also for broader food categories, rather than limited to specific substances. Specifically, in the present study, individual sensitivity to vanilla odour was negatively linked to consumption of sweet snacks in general, rather than only vanilla-flavoured snacks. Overall, these findings suggest that olfactory sensitivities can play a prominent general role in shaping individual snack choices.

Intriguingly, the present study observes a positive relationship between odour and taste sensitivities for sensory-incongruent snack choices. Specifically, high sensitivities to sweet-associated smell and taste (i.e., vanilla and sweet) were linked to increased choices of *savoury* foods, whereas high sensitivities to methional (potato odour) led to increased preference for sweet snacks. A recent study by Han, Keast, and Roura [45] found that individuals with high sensitivity to sweetness consumed less sweet food and showed increased consumption of non-sweet foods. Their intriguing observations indicate that the sensory compensation effect might also re-orientate food choices towards foods containing incongruent attributes. Our study supports these findings for sweetness and is the first to observe a similar effect linked to olfactory sensitivities.

Notably, the observed sensory effects on both congruent and incongruent snacks intakes were detected with measures of energy intake rather than consumption weight. Specific analyses of energy density further confirmed that heightened sensitivities were linked with increased choices of congruent snacks with lower energy and incongruent snacks with higher energy, despite comparable consumption weight. These findings suggested that individual odour and taste sensitivities predominantly modulate snack consumption by choices rather than portion judgements. Consistently, previous studies found that individuals showed similar judgement processes for portion sizes but distinct processes for energy density [22,23]. In particular, portion size judgements were thought to be linked with perception of external cues, e.g., units of food [82,83]. By contrast, judgements of energy density have been shown to be dependent of internal cues. For instance, in a study by Oliver, Wardle [84], stressed emotional eaters chose energy-dense snacks over low-calorie snacks more than unstressed and non-emotional eaters. The present analyses suggest that the role of individual olfactory and gustatory sensitivities in eating behaviour rests on judgements of energy density. Future research should be focusing on unravelling the mechanism underpinning the relationship between chemosensory processing and energy judgements.

The present study has some limitations. Specifically, individual snack choices were assessed in morning sessions in order to exert better control of food exposure prior to the study; however, snacking behaviour in a natural setting would be more likely to happen in afternoons and/or evenings [85,86]. Additionally, the small sample size of a relatively homogenous group of participants should be considered a general caveat of the study [87,88]. Future studies are warranted to replicate the current findings with larger and more diverse samples.

The present study is the first to test for links between individual olfactory and gustatory sensitivities and snack intake. Our findings reveal new links between odour/taste sensitivities and snack choices and intake. Overall, these findings highlight the fundamental role of chemosensory sensitivities in individual snack choice and provide novel insights into inter-individual variabilities in snack choices associated with differential health risks. Novel information derived from the present study provides better understanding of the sensory role in food choices, opening new opportunities for developing sensory interventions or products.

## Figures and Tables

**Figure 1 foods-11-00799-f001:**
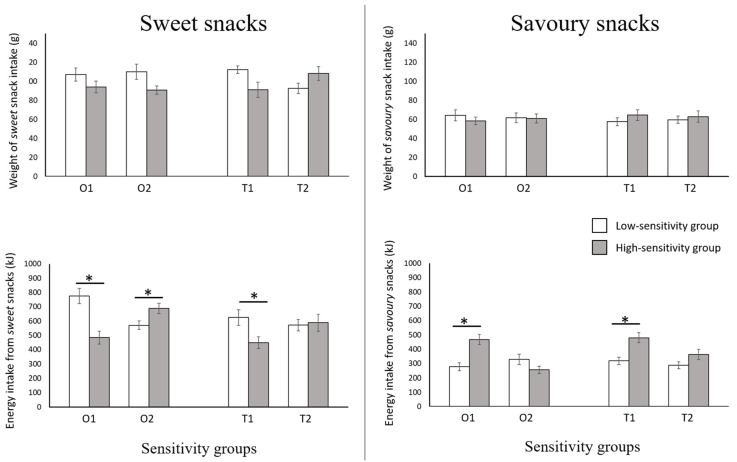
Bar graphs illustrating the weight and energy intake of *sweet* and *savoury* snack categories between “high” and “low” sensitivity groups, for two-olfactory (O1—vanillin, O2—methional) and two-gustatory stimuli (T1—sucrose, T2—NaCl). The error bars demonstrate the standard error for *sweet* and *savoury* snack intake measures. Significant differences (*p* < 0.05) in snack intake between the “high” and “low” sensitivity groups are indicated by an asterisk.

**Figure 2 foods-11-00799-f002:**
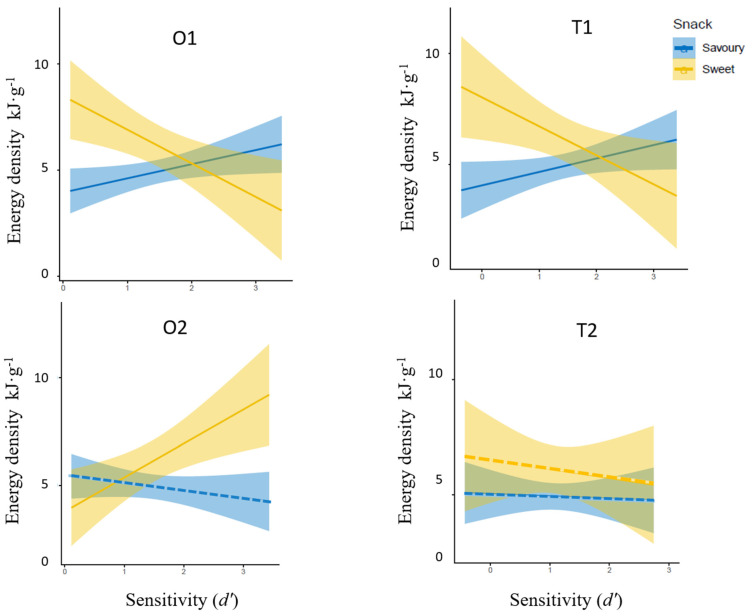
Correlations between sensitivity and energy density of *sweet* and *savoury* snack consumption for two-olfactory (O1—vanillin, O2—methional) and two-gustatory stimuli (T1—sucrose, T2—NaCl). Significant correlations were denoted by solid lines and non-significant correlations by dashed lines.

**Table 1 foods-11-00799-t001:** Summary of the characteristics of the stimuli used in the study.

Sensory Modality	Code	Chemical Name	Supplier	Descriptor	Concentration Range	Reference Concentration/Log Step
**Olfactory**	O1	4-Hydroxy-3-methoxybenzaldehyde	Vanesse, Camlin Fine Sciences, Mumbai, India	Vanillin; *Sweet-eliciting odour*	0.657 to 5.000 g·L^−1^	0.395 g·L^−1^ 0.221 log
	O2	3-(Methylthio) propionaldehyde	Sigma-Aldrich, Burlington, MA, USA	Methional Potato odour; *Savoury-eliciting odour*	0.260 to 1.000 μL·L^−1^	0.186 μL·L^−1^ 0.146 log
**Gustatory**	T1	Sucrose	Chelsea, Auckland, New Zealand	Sweet	9.537 to 15.100 g·L^−1^	8.5 g·L^−1^ 0.050 log
	T2	Salt (NaCl)	Labochem International, Frankfurt, Germany	Salty/savoury	1.683 to 2.667 g·L^−1^	1.5 g·L^−1^ 0.050 log

**Table 2 foods-11-00799-t002:** Summary of the snack items used in the food buffet. Respective energy content (in kJ) of each snack item is reported with the sugar/salt content.

Snack Category	Snack Item	Brand Details	Energy Content (kJ per 100 g)	Sugar Content in g (per 100 g)
**Sweet**	Chocolate	Cadbury^®^ Old Gold Chocolate, Mondelez Pty Ltd., Auckland, New Zealand	2372	56
	Chocolate chip cookies	CookieTime^®^ Choco Chunk, CookieTime, Christchurch, New Zealand	2010	39
	Crunchy granola	Muerli O & G^®^ Blueberry and Coconut, Nestle, Auckland, New Zealand	1710	23
	Red apples (Royal gala)	Fresh Produce Group, Auckland, New Zealand	218	10
				**NaCl in mg (per 100 g)**
**Savoury**	Salted peanut	ETA^®^ foods, Auckland, New Zealand	2440	620
	Corn chips	Mexicano^®^ Tasty Salsa, Wellington, New Zealand	1777	525
	Rice crackers	Pekish^®^ sour cream and chives, Monde Nissin, Mulgrave, Australia	1775	430
	Steamed broccoli—with 0.05% *w*/*w* added salt	Fresh produce group, Auckland, New Zealand	142	261

**Table 3 foods-11-00799-t003:** Summary of participants’ characteristics and baseline hunger level prior to the food choice task.

	Mean ± Standard Deviation	Range
**Age (years)**	26 ± 6	21–39
**BMI (kg∙m^−2^)**	27.1 ± 5.1	20.5–40.5
**DEBQ * score**		
** Restrained eating**	2.2 ± 0.7	1.0–3.9
** Emotional eating**	2.3 ± 0.9	0.8–4.7
** External eating**	3.3 ± 0.6	2.0–4.7
**Baseline hunger level**	20.2 ± 11.3	3.1–50.4

* DEBQ—Dutch Eating Behaviour Questionnaire [58], which assesses individual tendencies for restrained (1—unrestrained; 5—restrained eaters), emotional (1—unemotional; 5—emotional eaters), and external (1—non-external; 5—external eaters) eating behaviour.

**Table 4 foods-11-00799-t004:** Descriptive statistics of average *sweet* and *savoury* snack intake data in weight, weight percentage, energy, and energy density measures. Data are reported as mean ± standard deviation.

Snack Category	Weight (in Grams)	Weight Percentage	Energy (in kJ)	Energy Density (kJ·g^−1^)
**Sweet**	99.7 ± 56.7	61.1 ± 22.5	610 ± 426	6.1 ± 4.6
**Savoury**	63.7 ± 46.6	38.9 ± 22.5	311 ± 278	5.5 ± 4.6
** *t* ** **-statistic**	−4.07	−5.82	−4.88	−0.90
** *p* ** **-value**	*p* < 0.001	*p* < 0.001	*p* < 0.001	*p* = 0.368

## Data Availability

Data have been deposited in repository in Open Science Framework, OSF; https://osf.io/tq3ew/ (data has been deposited on 8 February 2022).

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
