# Peer review of "Olfactory and Gustatory Supra-Threshold Sensitivities Are Linked to Ad Libitum Snack Choice"

_foods, 2022, doi:10.3390/foods11060799_

Round 1
Reviewer 1 Report
- Although the authors adequately referenced many studies on the sensory perception of snacks, they failed to clearly indicate the novelty of the study in the Introduction section.
- The authors are recommended to highlight the industrial significance and potential application of the research findings, which might generate interest and effectively indicate contribution in the field.
Reviewer 2 Report
The overall aim of the research is clearly stated.
The aims and the results of the data are clearly and concisely stated in the abstract.
The introduction provides sufficient background information to enable readers to better understand the problem being identified by the Authors, but the introduction is too long, so I would suggest a reduction.
The data presented are of high quality and has been analyzed correctly.
Tables help the reader better understand the manuscript, but need to correct line 143
143 stimuli are presented in Table 1Error! Reference source not found.
Reviewer 3 Report
Foods
1612437
Olfactory and gustatory supra-threshold sensitivities are linked to ad libitum snack choice
Dear authors,
The present study aims to test for relationships of individual olfactory and gustatory sensitivities and their snack choices, in a laboratory-based ad libitum setting. The topic is good and the manuscript has been well written. The only thing is the authors should improve discussion section of the manuscript a little bit. My specific comment and questions are below;
- Line 66: What is BMI (boddy mass index)? Give the full name in the first place in the text!
- The authors should give some information about ad libitum in the introduction section!
- Line 143: Error?
- Line 184: Give more information about the equation!
- Line 223: their?
- Line 231: Standard deviations are too high! What could be the possible reason for that?
- Line 251: Standard error or standard deviation?
